# Cough dynamics in adults receiving tuberculosis treatment

Gwenyth O. Lee[ID][1,2]*, Germán Comina[1], Gustavo Hernandez-Cordova[1], Nehal Naik[ID][3], Oscar Gayoso[4], Eduardo Ticona[5,6], Jorge Coronel[7], Carlton A. Evans[8,9,10], Mirko Zimic[7,11], Valerie A. Paz-Soldan[1,8], Robert H. Gilman[7,8,12], Richard Oberhelman[1]

1 Department of Global Community Health and Behavioral Sciences, Tulane University, New Orleans, Louisiana, United States of America, 2 Department of Epidemiology, University of Michigan, Ann Arbor, Michigan, United States of America, 3 School of Medicine, Virginia Commonwealth University, Richmond, Virginia, United States of America, 4 Pulmonology Department, Hospital Nacional Cayetano Heredia, Lima, Peru, 5 Facultad de Medicina, Universidad Nacional Mayor de San Marcos, San Marcos, Peru, 6 Servicio de Enfermedades Infecciosas y Tropicales, Hospital Nacional Dos de Mayo Lima, Mayo, Lima, Perú, 7 Laboratorio de Investigación en Enfermedades Infecciosas, Laboratorio de Investigación y Desarrollo, Facultad de Ciencias y Filosofía, Universidad Peruana Cayetano Heredia, Lima, Peru, 8 Asociación Benéfica PRISMA, Lima, Perú, 9 Laboratory of Research and Development, Innovation For Health And Development (IFHAD), Facultad de Ciencias y Filosofía, Universidad Peruana Cayetano Heredia, Lima, Peru, 10 Department of Infectious Disease, Imperial College London, London, United Kingdom, 11 Laboratorio de Bioinformática y Biología Molecular, Facultad de Ciencias y Filosofía, Universidad Peruana Cayetano Heredia, Lima, Peru, 12 Department of International Health, Program in Global Disease Epidemiology and Control, Bloomberg School of Public Health, Johns Hopkins University, Baltimore, Maryland, United States of America

* golee@umich.edu

**Data Availability Statement:** The data has been made available in an open public data repository located at http://www.ifhad.org/data-repository/.

**Funding:** This work was supported from the National Institutes of Health (NIH), through Grant

## Abstract

Cough is a characteristic symptom of tuberculosis, is the main cause of transmission, and is used to assess treatment response. We aimed to identify the best measure of cough severity and characterize changes during initial tuberculosis therapy. We conducted a prospective cohort of recently diagnosed ambulatory adult patients with pulmonary tuberculosis in two tertiary hospitals in Lima, Peru. Pre-treatment and five times during the first two months of treatment, a vibrometer was used to capture 4-hour recordings of involuntary cough. A total of 358 recordings from 69 participants were analyzed using a computer algorithm. Total time spent coughing (seconds per hour) was a better predictor of microbiologic indicators of disease severity and treatment response than the frequency of cough episodes or cough power. Patients with prior tuberculosis tended to cough more than patients without prior tuberculosis, and patients with tuberculosis and diabetes coughed more than patients without diabetes co-morbidity. Cough characteristics were similar regardless of HIV co-infection and for drug-susceptible versus drug-resistant tuberculosis. Tuberculosis treatment response may be meaningfully assessed by objectively monitoring the time spent coughing. This measure demonstrated that cough was increased in patients with TB recurrence or co-morbid diabetes, but not because of drug resistance or HIV co-infection.

## Introduction

Cough is among the most characteristic symptoms of pulmonary tuberculosis. From the patient perspective, cough is a driver of care-seeking [1], and may significantly impact quality

4D43TW009349-05. Nehal Naik was supported by the National Institutes of Health Office of the Director, Fogarty International Center, Office of AIDS Research, National Cancer Center, National Heart, Blood, and Lung Institute, and the NIH Office of Research for Women's Health through the Fogarty Global Health Fellows Program Consortium comprised of the University of North Carolina, John Hopkins, Morehouse and Tulane (R25TW009340). CAE acknowledges funding from the Wellcome Trust (awards 057434/Z/99/Z, 070005/Z/02/Z, 078340/Z/05/Z, 105788/Z/14/Z and 201251/Z/16/Z); the Joint Global Health Trials consortium (MRC, DFID, & Wellcome Trust award MR/K007467/1); the STOP TB partnership's TB REACH initiative funded by the Government of Canada and the Bill & Melinda Gates Foundation (awards W5_PER_CDT1_PRISMA and OPP1118545); and the charity IFHAD: Innovation For Health And Development. The content is solely the responsibility of the authors and does not necessarily represent the official views of the NIH or any of the funders.

**Competing interests:** The authors have declared that no competing interests exist.

of life [2], both physically and psychologically. Cough is also associated with fear of infection, stigma, and the social isolation of patients with tuberculosis [3]. From the clinical perspective, cough is critical both in diagnosis [4] and to monitor treatment response. Recently, it has been shown that cough frequency is associated with sputum bacillary load [5], and with the extent of cavitary lung disease [6]. Cough is also uniquely relevant for the transmission of tuberculosis [7]. For instance, it has been suggested that bouts of acute cough associated with common respiratory pathogens may drive transmission from otherwise subclinical tuberculosis disease [8]. Cough frequencies may also be associated with tuberculin conversion rates in patients' household contacts [9].

Despite the major role that cough plays in tuberculosis transmission, clinical management, and patient experience, relatively few studies have objectively described cough among patients with tuberculosis, in part due to logistical challenges in measuring and characterizing cough. Recently, we reported the development of an audio-based device [10–12] to measure cough frequency among HIV-negative Peruvian patients with drug-susceptible tuberculosis disease [5,6,13]. Although this device is non-invasive and capable of generating 24-hour cough recordings, the approach had some limitations. Thirty-seven percent of recordings were excluded for technical reasons, primarily background noise [5]. In addition, because the detection of cough was based on sound, varying levels of background noise make it more difficult to comparably extract features of cough beyond frequency, such as cough intensity. Furthermore, the collection of audio data created a risk to patient privacy. Although the recording was mostly processed algorithmically, human review of segments to confirm a computer-algorithm identified potential cough was still required to ensure accuracy [14].

The present study describes the application of an adapted CayCaMo cough monitor using a solid-state piezoelectric vibration-sensing device (vibrometer) to:

1. Describe features of cough severity among patients with tuberculosis pre-treatment and describe cough treatment response.

2. Determine which feature is most predictive of objective microbiological measures of TB severity before and during TB treatment.

3. As an exploratory analysis, to identify possible determinants of cough severity, including HIV sero-status, diabetes co-infection, and tuberculosis drug resistance. Given that the management of HIV co-infection and multi-drug resistance are two major current challenges to global tuberculosis control,[15] understanding the dynamics of cough among these patient groups has important implications for transmission.

## Materials and methods

### Cough monitor description

Cough recordings were collected using a modified version of our existing cough recording device (Fig 1) [16]. The modification consisted of replacing the internal electric microphone by a solid-state piezoelectric speaker (SWT, Part No: 3B27+3.9EA, 27 mm diameter, resonance frequency 3.9±0.5 KHz) that creates a variable electric charge based on vibration. The utility of this device is that the response spectrum of the sensor is such that sounds with frequencies beyond 4Khz are highly attenuated. As a result, the device is capable of recording cough sounds with high accuracy, while the spoken word is unintelligible. Furthermore, because the sensor is placed on the suprasternal notch of the patient and detects vibration, features of cough such as cough intensity or duration can be detected with greater sensitivity than by

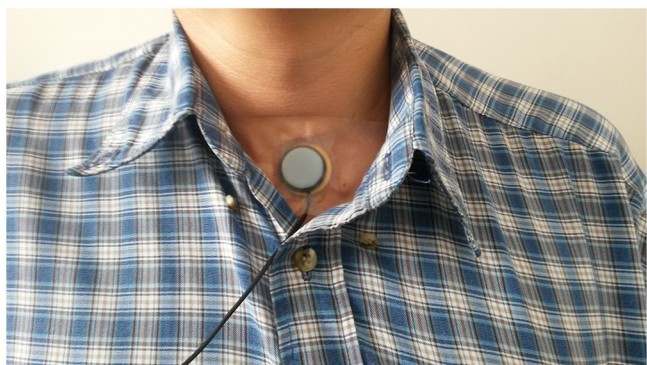

**Fig 1. Piezoelectric sensor.** Shown here is the vibrometer, as it would be worn by a participant.

audio alone. Our previously reported algorithm then used identify potential coughs based on the sensor signal, with a sensitivity of 75.5% and a Birring specificity of 99.3% among adults. [10,11]. These recordings were then reviewed by a human listener to further increase sensitivity [14]. In a set of test recordings in which induced cough and non-cough sounds (throat clearing and spoken words) were captured by both audio and the vibration-based sensor, there was perfect agreement between classification of sounds between the two methods. A full description of this validation of the modified device will be reported elsewhere.

## Study description

From June 2016 to March 2017, we prospectively followed a cohort of 71 ambulatory adult (aged $\geq$18 years) patients with tuberculosis disease in two tertiary hospitals in the city of Lima, Peru: Hospital Dos de Mayo (DM) and Hospital Cayetano Heredia (CH). The sample size was determined based on a calculation to detect differences in the proportion of positive microbiologic results between patients with and without cough and the study was not designed, *a priori*, to detect differences in cough between patients based on HIV co-infection or drug resistant tuberculosis.

Newly identified adult (at least 18 years old) patients diagnosed by the Peruvian health care system by the presence of at least once acid-fast bacilli positive sputum smear microscopy test were eligible to participate in the study (henceforth referred to as smear). At enrollment, patients were assisted to complete a questionnaire detailing prior history of tuberculosis, the presence of co-morbidities, and socio-economic position. The latter was characterized using the progress out of poverty index (PPI), a non-income-based wealth index developed for Peru [17,18]. Values for the PPI range from 0–100.

Six cough recordings were made for each patient during the first 60 days of treatment. The first of these was done, when possible, on day 0 (the day of diagnosis) and additional visits were scheduled after 3, 7, 14, 30, and 60 days of treatment. Based on previous findings that a 4-hour cough recording accurately approximated cough frequency over 24-hours [5], recordings were 4 hours in duration. Based on prior evidence that the frequency of cough varied throughout the day, peaking from 1–2 pm [5], whenever possible recordings were scheduled then.

Smear tests were performed by the Peruvian health system, and additional smear and microscopic-observation drug susceptibility (MODS) tests incorporating drug-susceptibility testing for isoniazid and rifampicin [19–21] were performed by the study team. All results were communicated promptly to the participant's medical team. Patients underwent treatment

according to the current Peruvian national guidelines [22], using direct observation of every treatment dose and this study had no role in patient care or treatment.

The study was approved in Peru by the institutional review boards of the Universidad Peruana Cayetano Heredia, the Asociación Benéfica PRISMA, and both participating Peruvian hospitals. In the USA, it was reviewed by the Johns Hopkins School of Public Health and the Tulane University School of Public Health and Tropical Medicine through an inter-institutional authorization agreement.

## Cough quantification

Cough events were summarized as in our previous studies into 'episodes' (also termed 'coughing fits' or 'epochs') defined as any series of coughs separated by <2 seconds between each cough. For each episode, we calculated duration, peak energy or amplitude, and power (spectral power summed over the duration of the episode) using spatial analysis (Fig 2). These features were summarized across all episodes detected over the 4-hour recording in two ways. First, the *per-episode geometric mean*, indicators of the strength of the typical cough episode, were calculated as the average episode: DURATION; AMPLITUDE; and POWER. Additionally, the *hourly sum* of episodes' total: TIME; and POWER together with the COUGH EPISODE FREQUENCY per hour were calculated. Thus, our parameters were:

1. AVERAGE EPISODE DURATION (seconds)
2. AVERAGE EPISODE PEAK AMPLITUDE (seconds)
3. AVERAGE EPISODE POWER (milliwatts)
4. TOTAL TIME COUGHING per hour (seconds per hour)
5. TOTAL POWER EXPENDED COUGHING (milliwatts per hour)
6. COUGH EPISODE FREQUENCY (episodes per hour)

## Statistical methods

To describe cough severity, we calculated Spearman correlations to examine the relationship between features. Interclass correlation coefficients were estimated to determine the degree to which variability between features was explained by within-participant versus between-

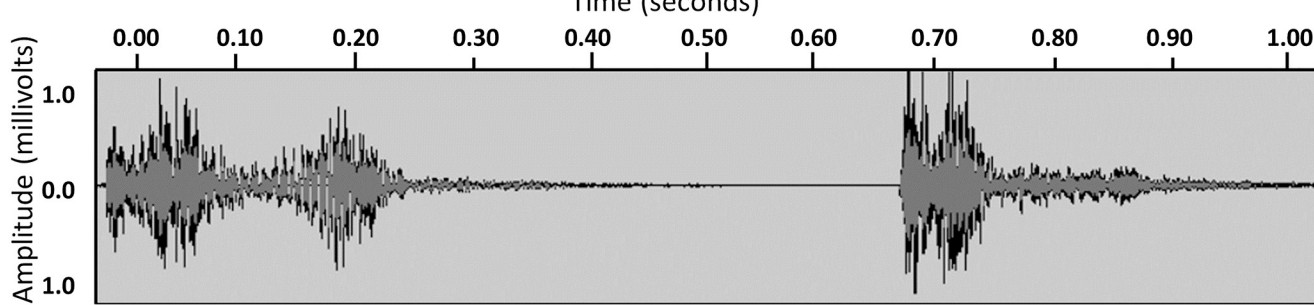

**Fig 2. Cough signal.** Shown here are two individual coughs. Because these coughs took place less than two seconds apart, they would be classified as part of a single episode, with a total duration of 1.00 second. The episode PEAK is the maximum amplitude over the episode (max(abs(signal))), and the episode POWER is the rms(signal)^2.

participant variability. To characterize cough response to treatment, we visualized the dynamics of each cough feature over time. As previously, we defined any cough frequency of 0.7 confirmed cough events per hour or fewer as normal for a healthy adult [5,23,24].

To determine which feature of cough best predicted objective microbiological measures of tuberculosis severity, we developed bivariable models to examine the relationship between each feature and microbiologic results from its paired sputum sample. We treated time to MODS positivity (TTP) as our primary outcome of interest, and, secondarily, considered outcomes of MODS positivity and sputum smear positivity (including paucibacillary results). We compared the fit of these models by their log-likelihood and by Akaike's Information criterion (AIC), a penalized-likelihood criteria for comparing non-nested models [25]. Using the same methodology and among the subset of participants with completed pre-treatment and 3-day recordings, we examined the relationship between the percent change in each feature from day 0 to day 3 and outcomes of MODS positivity and TTP. To identify the optimal cutoff value for cough features versus MODS positivity, we also ran models stratified by the day of treatment and constructed ROC curves based on the model output.

To identify determinants of cough severity, we constructed bivariable and multivariable models to examine whether features of cough episode severity varied by HIV serostatus, presence of mono or multidrug-resistant tuberculosis, or concurrent diabetes. We used Tobit models to model the relationship between these clinical factors and AVERAGE EPISODE DURATION and negative binomial models to model the relationship between these clinical factors and EPISODE FREQUENCY. Tobit models yield unbiased estimates when the dependent variable is censored [26], and were used here to account for recordings where no cough was reported. The lower limit for censoring was taken as the lowest observed value of each cough feature.

Because we wished to test the association between HIV status, drug resistant disease, and cough, these variables were retained in final multivariable models regardless of statistical significance. Other potentially confounding variables were included in the multivariable model based on comparisons of Akaike's criterion for final variable selection. To ensure that associations between HIV, drug-resistant tuberculosis, and cough were not confounded by prior tuberculosis or treatment, models were also run where all participants with prior tuberculosis were excluded, and for patients on first-line treatment only, as a sensitivity analysis. We considered adjusting for the following: day of treatment, patient age, sex, HIV status, diagnosis of diabetes, history of smoking, and history of prior tuberculosis, the patient's microbiological result (MODS positive or negative) at the same visit, and whether the patient's tuberculosis was drug resistant or not.

All analyses were completed in MATLAB (The MathWorks, Massachusetts, USA) or using STATA statistical software version 15.1 (StataCorp LP, College Station, Texas, USA).

## Results

### Participant characteristics

Seventy-one patients were enrolled. Sixty-nine provided at least one successful recording and were considered 'analyzable' cases. A total of 363 recordings were collected, of which one was unusable. 43 participants had complete data (6 recordings), 15 had 5 complete recordings, and 11 had 4 or fewer completed recordings. Among completed recordings, 358 were paired with a MODS test result from the same visit (S1 Fig). Characteristics of the analyzable cases are shown in Table 1. All participants living with HIV had been previously diagnosed and were receiving antiretroviral therapy; no additional HIV-associated comorbidities, such as *Pneumocystis carenii* pneumonia, were documented. Fifty-two recordings were available from patients

**Table 1. Participant characteristics.**

| Characteristic | (N = 69) | Percent / Mean (SD) |
|---|---|---|
| Percent Male | 41 | 59% |
| Progress out of Poverty Index (PPI) | n/a | 58 (12) |
| Drug sensitive | 59 | 86% |
| Rifampicin-resistance only | 0 | 0% |
| Isoniazid resistance only | 2 | 3% |
| Multi-drug resistance | 8 | 12% |
| HIV positive | 8 | 12% |
| HIV positive and drug resistant TB | 1 | 2% |
| Diabetes | 6 | 9% |
| Prior diagnosis of TB | 7 | 10% |
| Prior diagnosis of TB and drug resistance | 3 | 4% |
| Smoker | 13 | 19% |

who had not yet started therapy (i.e. on their day of diagnosis). The final study sample includes 69 patients with at least one paired cough recording and microbiological result., 6 of whom had diabetes co-morbidity and 7 of whom had previous TB disease.

By treatment day 14, 1 patient had confirmed drug-resistant tuberculosis and had been changed to second-line therapy, while 7 patients with drug-resistant disease that was later confirmed, were still on first-line therapy. By day 14, 48% of patients had clinically normal cough rates. This was similar for patients with drug-sensitive (46%) and drug-resistant disease (63%).

## Features of cough pre-treatment and cough treatment response

EPISODE FREQUENCY was correlated with TOTAL TIME COUGHING and TOTAL POWER EXPENDED COUGHING (rho = 0.96, p<0.001 and rho = 0.67, p<0.001), and was not associated with AVERAGE EPISODE DURATION, PEAK AMPLITUDE, or POWER. AVERAGE EPISODE PEAK AMPLITUDE and POWER were strongly (rho>0.6) correlated with each other (S1 Table). Ten percent to 29% of the variability in each cough episode feature was explained by within-individual variability (S2 Table).

One hundred and fourteen recordings (32%) had no coughs recorded over the 4-hour period. Of the 52 patients with pre-treatment cough recordings, 11 patients (21%) had a pre-treatment recording with a cough frequency similar to that of a healthy adult (< = 0.7 cough events/hour), and 9 (17%) did not cough during their 4-hour pre-treatment recording. All features were right-skewed, such that many participants had relatively mild cough (low frequency and short duration) and a small proportion had substantially more severe cough. At baseline, patients at the 75th percentile had an EPISODE FREQUENCY 9.4 episodes per hour, equivalent to 11 times that of patients at the 25th percentile (0.89 episodes per hour); a TOTAL TIME COUGHING of 12 seconds per hour (22 times more than patients at the 25th percentile, who coughed 0.56 seconds per hour), and a TOTAL POWER EXPENDED 48 times greater than patients at the 25th percentile (0.60 versus 0.013 milli-watts per hour). EPISODE FREQUENCY, TOTAL TIME COUGHING, and TOTAL POWER EXPENDED each decreased with treatment, as did AVERAGE EPISODE DURA-TION. In contrast, AVERAGE EPISODE PEAK AMPLITUDE and POWER remained approximately stable over time (S2 Fig).

## Association with objective microbiological measures of tuberculosis severity

EPISODE FREQUENCY, TOTAL TIME COUGHING, and AVERAGE EPISODE DURA-TION were statistically significantly associated with TTP, as well as MODS positivity. EPISODE FREQUENCY, TOTAL TIME COUGHING, TOTAL POWER EXPENDED COUGHING, and AVERAGE EPISODE DURATION were each statistically significantly associated with smear positivity. AVERAGE EPISODE PEAK AMPLITUDE and POWER were not associated with microbiologic results. Based on model fit, TOTAL TIME COUGH-ING was the strongest predictor of TTP (AIC = 1197), followed by COUGH EPISODE FRE-QUENCY (AIC = 1200) and AVERAGE EPISODE DURATION (AIC = 1205). TOTAL TIME COUGHING was also the best predictor of MODS positivity and smear positivity (S3 Table). Decreases in Akaike information criterion of four or greater have been described as "signifi-cant" [27]; using this guideline TOTAL TIME COUGHING was the best cough measure com-pared to EPISODE FREQUENCY. There was no evidence that early changes in any cough feature (% change from day 0 to day 3) were associated with smear, MODS, or TTP results (model results not shown). In stratified models, TOTAL TIME COUGHING was predictive of MODS positivity on day 0 of treatment (AUC = 0.73, 95% CI: 0.42, 01.00, S3 Fig). However, stratified analyses at other time points (day 3 to 60) suggested that features of cough were not predictive of MODS positivity (results not shown), rather, only in combined analyses (all treat-ment days combined) were features of cough significantly associated with MODS results.

## Determinants of cough severity

Because TOTAL TIME COUGHING was the strongest predictor of TTP, this feature was car-ried forward to examine determinants of cough severity. Secondarily, because we have previ-ously reported on the association between COUGH EPISODE FREQUENCY and patient characteristics in a separate cohort of HIV-negative Peruvian patients with drug-sensitive tuberculosis [5,6,13], bivariable and multivariable incidence rate ratios comparing COUGH EPISODE FREQUENCY based on clinical characteristics from treatment day 0 to 60 (n = 357) are also reported.

TOTAL TIME COUGHING was significantly positively associated with: diabetes ($\beta$ = 0.86, 95% CI: 0.09, 1.63, p = 0.028); history of prior tuberculosis ($\beta$ = 1.44, 95% CI: 0.66, 2.22, p<0.001) but not HIV status nor drug resistant tuberculosis. Smokers with TB also tended to cough more than non-smokers with TB, although this was not statistically significant. COUGH EPISODE FREQUENCY was significantly associated with the same factors (Table 2 and Fig 3).

## Discussion

Using our new device, we document spontaneous cough frequency and severity. Our results corroborate previous findings that cough frequency is associated with positive sputum culture for *M. tuberculosis* [5,9]. We further find that the average duration of cough episodes was asso-ciated with microbiologic positivity, and that the hourly cough duration (a feature that com-bines cough episode frequency and per-episode duration) is more predictive of MODS TTP than either feature individually. In contrast, the average peak and power of cough episodes were not related to microbiology and did not change over time.

Cough intensity can be measured via audio [28], however, this approach has limited feasi-bility in real-world, out-patient scenarios. A vibrometer-based approach surmounts these obstacles and allows additional features like cough episode peak and power to be reliably

**Table 2. Determinants of cough severity.**

| | Ln (TOTAL TIME COUGHING) | | EPISODE FREQUENCY | |
|---|---|---|---|---|
| | Bivariable | Multivariable | Bivariable | Multivariable |
| | Beta Coefficient (95% Confidence Interval) | Beta Coefficient (95% Confidence Interval) | Rate Ratio (95% Confidence Interval) | Rate Ratio (95% Confidence Interval) |
| | (p-value) | (p-value) | (p-value) | (p-value) |
| **Treatment day** | -0.66 (-0.91, -0.40) | -0.66 (-0.92, -0.41) | 0.66 (0.52, 0.84) | 0.61 (0.46, 0.79) |
| | (p<0.001) | (p<0.001) | (p = 0.001) | (p<0.001) |
| **Treatment day ^2** | 0.06 (0.02, 0.10) | 0.06 (0.02, 0.10) | 1.03 (1.00, 1.06) | 1.04 (1.00, 1.07) |
| | (p = 0.002) | (p = 0.001) | (p = 0.063) | (p = 0.028) |
| **Age*** | 0.11 (-0.26, 0.47) | n/a | 0.98 (0.82, 1.16) | n/a |
| | (p = 0.575) | | (p = 0.785) | |
| **Sex = Female** | -0.05 (-0.59, 0.49) | n/a | 1.05 (0.63, 1.75) | n/a |
| | (p = 0.843) | | (p = 0.846) | |
| **Smoker** | 0.98 (-0.21, 2.17) | 0.59 (-0.06, 1.25) | 2.01 (1.34, 3.01) | 1.82 (0.90, 3.73) |
| | (p = 0.108) | (p = 0.077) | (p = 0.001) | (p = 0.094) |
| **MODS positive** | 0.71 (0.31, 1.10) | n/a | 1.73 (1.14, 2.63) | n/a |
| | (p<0.001) | | (p = 0.010) | |
| **Prior TB** | 1.04 (0.27, 1.82) | 1.44 (0.66, 2.22) | 2.66 (1.51, 4.66) | 4.03 (2.19, 7.42) |
| | (p = 0.009) | (p<0.001) | (p = 0.001) | (p<0.001) |
| **Diabetes** | 0.59 (-0.24, 1.43) | 0.86 (0.09, 1.63) | 1.38 (0.81, 2.37) | 1.75 (1.13, 2.71) |
| | (p = 0.164) | (p = 0.028) | (p = 0.240) | (p = 0.012) |
| **HIV positive** | 0.03 (-0.83, 0.88) | 0.56 (-0.21, 1.34) | 0.73 (0.40, 1.33) | 1.40 (0.71, 2.78) |
| | (p = 0.952) | (p = 0.150) | (p = 0.305) | (p = 0.366) |
| **Drug resistant TB** | -0.03 (-0.83, 0.77) | -0.35 (-1.09, 0.39) | 1.33 (0.45, 2.77) | 0.73 (0.37, 1.45) |
| | (p = 0.942) | (p = 0.351) | (p = 0.784) | (p = 0.366) |

Shown here are the results of bivariable and multivariable Tobit regression models in which the outcome of interest is the log-transformed TOTAL TIME COUGHING (log seconds per hour). The lowest observed value was taken as the lower limit of the model. This approach allows recordings where no cough episodes were observed to be included in the analysis. To model COUGH EPISODE FREQUENCY, bivariable and multivariable negative binomial regression models were constructed. Both models included a random intercept to account for the correlation between recordings from the same study participant. Microbiology (MODS result) was collinear with treatment day (the relationship between characteristics of cough and treatment day is explained by microbiological response), therefore the final multivariable includes treatment day and not MODS result. Treatment day was adjusted for in the model using a quadratic term (treatment day ^2) to reflect the non-linear, rapid decrease in cough observed early in treatment [5].

*Age per 10 years, centered at 34 years.

(Total participants = 69, total recordings = 359)

captured. It has recently been shown that, for voluntary cough, sound power and sound energy correspond to patient self-reported cough strength as well as cough flow and pressure [28].

As in our previous study, we found significant heterogeneity in cough between participants, both pre-treatment and during treatment. This may limit the utility of objective cough monitoring as a clinical tool. Although individuals with greater cough were, on average, more likely to be MODS positive, there was not a sensitive and specific threshold for cough frequency that consistently predicted a positive sputum culture. Our results also demonstrate that a proportion of patients with pulmonary tuberculosis have low cough frequencies, and brief cough duration, even prior to treatment [5]. Better understanding the factors that drive some individuals to cough much more often, and more severely, than others, may be useful to better understanding tuberculosis transmission dynamics. We also found no evidence that early changes in cough dynamics, such as decreases in episode frequency, duration, or power after 3 days of treatment, are associated with longer-term microbiological positivity.

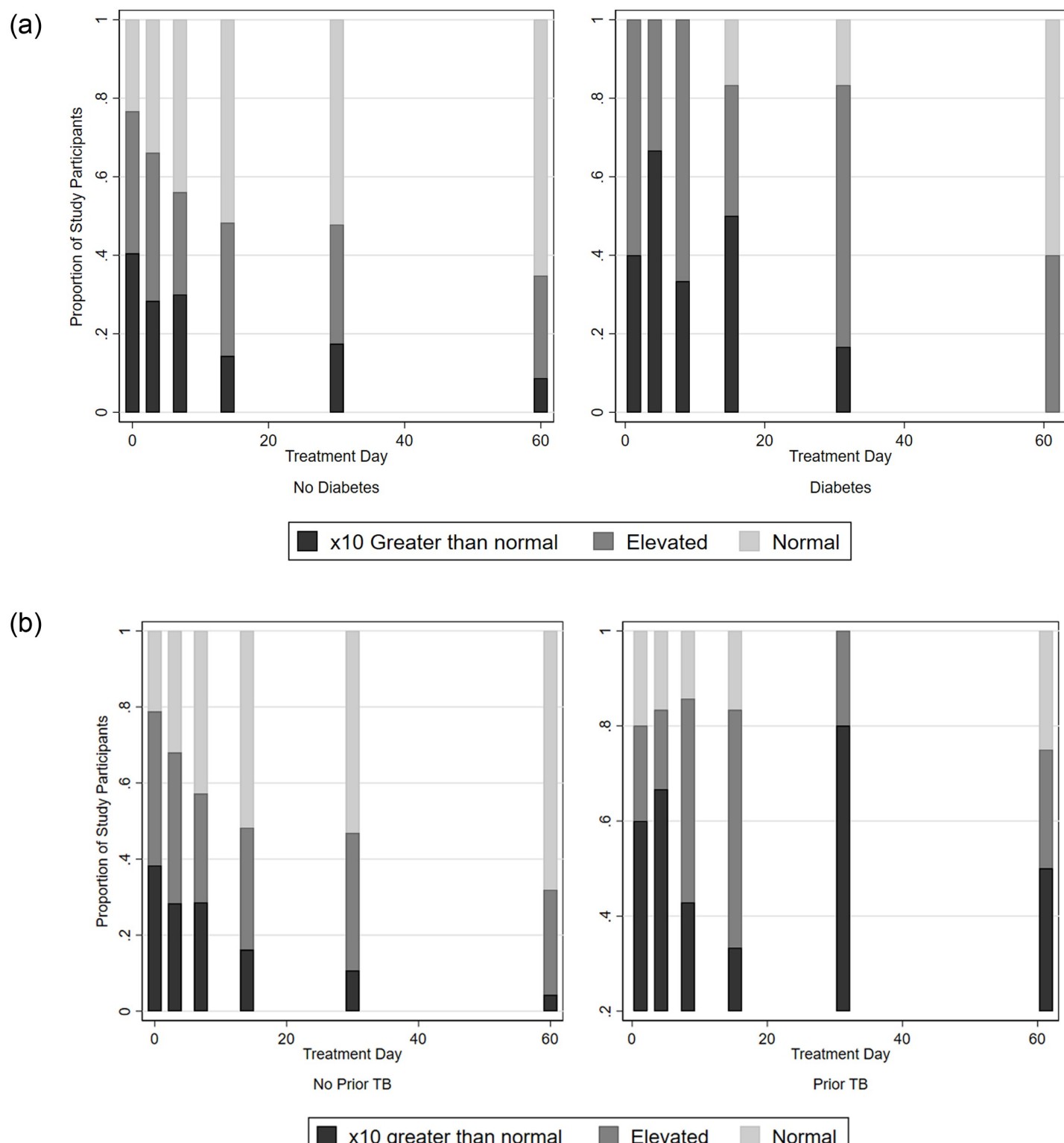

**Fig 3. Total time spent coughing by prior tuberculosis diagnosis and diabetes co-morbidity.** Shown here are stacked bar graphs showing the proportion of individuals in each group with extremely elevated (x10 greater than normal), elevated (greater than normal), or normal ($< = 0.6$ coughs/hour) cough as of a given study visit. Individuals with prior TB and individuals with diabetes co-infection were more likely to have extremely elevated or elevated cough at later study visits.

Voluntary cough power may differ by gender, height, and lung function and there is a need to better understand how other factors, such as ethnicity and smoking status affect cough frequency and intensity [28]. Given the small number of participants living with HIV, with diabetes co-infection, with a drug resistant infection, or presenting with other risk factors, the results we present here should be regarded as exploratory. We found that smoking status was associated with time spent coughing, a finding also supported by other studies [9]. COPD is a common occurrence in smokers, and while COPD was not noted in the clinical records of any patient, it is possible that this was an unmeasured, confounding variable. Patients with prior tuberculosis also coughed more. This may be due to more extensive lung damage in these patients [6]. or physiological alterations of the lung that did not fully recover to normality. Finally, we found that patients with concurrent diabetes coughed more than their non-diabetic counterparts, which is consistent with recent reports of more severe tuberculosis disease in patients with diabetes [29]. Future studies with larger samples of patients with diabetes would be useful to confirm these preliminary observations.

We found no evidence of association between HIV coinfection, or drug resistant tuberculosis and cough. In theory, patients with HIV might be expected to have milder cough due to suppressed inflammatory responses, but our data did not support this. Patients with drug-resistant tuberculosis began with cough rates like those of patients with drug-sensitive tuberculosis, and, by day 14 of treatment, a similar proportion had achieved clinically normal cough, although most (7/8) were still on first-line treatment. This is likely explained by most MDR-tuberculosis strains being susceptible to at least one of the first-line drugs administered empirically. Therefore, the utility of cough monitoring for the early prediction of treatment failure may be limited.

No cough was detected in a third of recordings. This creates considerations for analysis that are similar to those observed in biomarker data, where lower limits of detection can result in left-censored data [30]. In this study, we used Tobit regression models to account for this in instances where cough features were the independent variable of interest, and limited analyses where cough features were the dependent variables of interest to complete-case analysis.

Our vibration-based method of cough recording has benefits over previous audio- based methods [5,13]. Privacy for the patient is improved, as the device accurately detects vibrations in the frequency produced by cough, but not in the frequency produced by human speech. The device is unobtrusive and can easily be hidden with clothing without interfering with the signal quality. Finally, because the vibrometer is not influenced by ambient noise, cough intensity can be more reliably estimated and compared.

## Conclusions

Total hourly cough duration (seconds spent coughing per hour) was a better predictor of the microbiologic response to treatment than episode frequency and patients with prior tuberculosis had higher cough rates that patients without prior tuberculosis. Better understanding cough dynamics across populations and sub-groups may inform our understanding of TB transmission.

## Supporting information

**S1 Fig. Study flow chart.** Shown here are the total number of cough recordings and microbiologic testing completed over the course of the study. 357 cough recordings (from 69 study participants) were matched to a microbiology result.
(TIF)

**S2 Fig. Features of cough by treatment day.** Shown here are the percentages of study participants with an elevated cough rate, and the geometric means of other characteristics of cough

episodes, on specific days of treatment. By day 14 of treatment, 48% of patients had clinically normal cough rates. Recordings were taken as having occurred sufficiently near to the target day if they occurred strictly prior to the start or treatment, or on the first day of treatment (target day 0), or within +/-2 days (target day 3 and 7), +/- 7 days (target day 14), or +/-20 days (target days 30 and 60) of the target date.
(TIF)

**S3 Fig. ROC curve.** Shown here are receiver-operator curves (ROC) for MODS positivity versus TOTAL TIME COUGHING in 52 pre-treatment recordings (47 MODS positive and 5 MODS negative). AUC = 0.73; Best cut-off = 1.9 episodes/hour; Sensitivity = 85%; Specificity = 50%.
(TIF)

**S1 Table. Spearman correlation between cough features.** Shown here is a heat map representing non-parametric correlation coefficients between cough features, based on 43 of 52 pre-treatment cough recordings where at least one cough was recorded during the 4-hour recording. Results on treatment days 3–60 are similar (not shown).
(DOCX)

**S2 Table. Intraclass correlation coefficients.** Shown here are intraclass correlation coefficients for each cough feature, describing the proportion of variability in each feature explained by within-individual variability. For example, 29% of the variability in COUGH EPISODE FREQUENCY can be explained by within-individual variability.
(DOCX)

**S3 Table. a-c.** Association Between Cough Features and Microbiological Outcomes. These tables examine the extent to which specific features were predictive of microbiological outcomes. The primary microbiological outcome of interest was MODS time to positivity (TTP). Secondarily, we also considered the microbiological outcomes of MODS positivity (+/-) and smear positivity (+/-). All cough features are log-transformed (natural log AVERAGE EPISODE DURATION, natural log AVERAGE EPISODE PEAK AMPLITUDE, etc.). All models are bivariable models, unadjusted for treatment day or other factors, as it is expected that the relationship between characteristics of cough and treatment day was explained by microbiological response. All models include only recordings with at least one recorded cough episode included (complete case analysis). TTP models were Tobit models to account for the structure of the TTP data (TTP results 0 and 21 are treated as continuous, and TTP results of 22 or greater (equivalent to a negative MODS culture) are right-censored. MODS (+/-) and smear models were logistic models. All models included a random effect to account for within-patient variability. Log-likelihood (LL) and Akaike's information criterion (AIC) were compared between models with each feature as the independent variable, where the model with the lowest AIC suggests that this feature is the strongest individual predictor of smear positivity.
(DOCX)

## Acknowledgments

We would like to thank Karyna Liseth Quesada Torres, Evelin Yuliza Chuchon Quispe, Carmen Rosa Hilario Cardenas de Yap, Shirley Carol Venegas Morocho, and Hector Jesus Arteaga Pillaca for their invaluable help in data collection and sample processing. We also wish to thank the study participants for their time and support, and Jose Lopez and Alvaro Proaño for their support of the study.

## Author Contributions

**Conceptualization:** Gwenyth O. Lee, Germán Comina, Carlton A. Evans, Mirko Zimic, Valerie A. Paz-Soldan, Robert H. Gilman, Richard Oberhelman.

**Data curation:** Gwenyth O. Lee, Germán Comina, Nehal Naik.

**Formal analysis:** Gwenyth O. Lee, Germán Comina, Carlton A. Evans.

**Funding acquisition:** Nehal Naik, Carlton A. Evans, Mirko Zimic, Valerie A. Paz-Soldan, Robert H. Gilman, Richard Oberhelman.

**Investigation:** Gwenyth O. Lee, Germán Comina, Jorge Coronel, Richard Oberhelman.

**Methodology:** Gwenyth O. Lee, Germán Comina, Mirko Zimic, Valerie A. Paz-Soldan.

**Project administration:** Gwenyth O. Lee, Germán Comina, Gustavo Hernandez-Cordova, Oscar Gayoso, Eduardo Ticona, Mirko Zimic, Valerie A. Paz-Soldan, Robert H. Gilman, Richard Oberhelman.

**Resources:** Oscar Gayoso, Eduardo Ticona, Mirko Zimic, Valerie A. Paz-Soldan, Robert H. Gilman, Richard Oberhelman.

**Software:** Germán Comina, Mirko Zimic.

**Supervision:** Oscar Gayoso, Eduardo Ticona.

**Writing – original draft:** Gwenyth O. Lee.

**Writing – review & editing:** Gwenyth O. Lee, Germán Comina, Gustavo Hernandez-Cordova, Nehal Naik, Oscar Gayoso, Eduardo Ticona, Jorge Coronel, Carlton A. Evans, Mirko Zimic, Valerie A. Paz-Soldan, Robert H. Gilman, Richard Oberhelman.

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
