## [Decision Letter · Decision Letter 0]

4 Feb 2020

PONE-D-19-31712

COUGH DYNAMICS IN ADULTS RECEIVING TUBERCULOSIS TREATMENT

PLOS ONE

Dear Dr. Lee,

Thank you for submitting your manuscript to PLOS ONE. After careful consideration, we feel that it has merit but does not fully meet PLOS ONE’s publication criteria as it currently stands. Therefore, we invite you to submit a revised version of the manuscript that addresses the points raised during the review process.

We would appreciate receiving your revised manuscript by Mar 20 2020 11:59PM. To enhance the reproducibility of your results, we recommend that if applicable you deposit your laboratory protocols in protocols.io, where a protocol can be assigned its own identifier (DOI) such that it can be cited independently in the future. For instructions see: http://journals.plos.org/plosone/s/submission-guidelines#loc-laboratory-protocols

We look forward to receiving your revised manuscript.

Kind regards,

HASNAIN SEYED EHTESHAM

Academic Editor

PLOS ONE

Additional Editor Comments (if provided):

Major Revision

Journal Requirements:

2. Please include a reference for the study mentioned at line 98-99, which illustrates the validation of the instrument used in the present study.

5. Please ensure that you refer to Figure 1 in your text as, if accepted, production will need this reference to link the reader to the figure.

Reviewers' comments:

Reviewer's Responses to Questions

**Comments to the Author**

1. Is the manuscript technically sound, and do the data support the conclusions?

Reviewer #1: Yes

Reviewer #2: Partly

2. Has the statistical analysis been performed appropriately and rigorously? 

Reviewer #1: I Don't Know

Reviewer #2: Yes

3. Have the authors made all data underlying the findings in their manuscript fully available?

Reviewer #1: Yes

Reviewer #2: Yes

4. Is the manuscript presented in an intelligible fashion and written in standard English?

Reviewer #1: Yes

Reviewer #2: Yes

5. Review Comments to the Author

Reviewer #1: It is a novel presentation in which a symptom which is subjective has been expressed objectively . A mention on the calibration of the test also the specificity and sensitivity of this test needs to be addressed.

Did the exclusion criteria list smokers ?

COPD is a common occurrence in smokers and can be a confounding variable for the present study.

HIV patients are mentioned . Although the complications of HIV patients also need to be addressed such as pneumocystis carenii pneumonia.

Reviewer #2: Manuscript #: PONE-D-19-31712

Title: COUGH DYNAMICS IN ADULTS RECEIVING TUBERCULOSIS

TREATMENT

Review:

The study is interesting, but the total number of subjects included in the study (n=69) is very small to reach any substantial conclusion. These numbers further reduced significantly in the sub groups, the researchers have chosen i.e. (Drug resistant n=8; HIV positive n=8; Diabetics n=6; smokers n=13) etc.

Hence, with such a small group of participants, the hypothesis, which the authors are putting forward, cannot be substantiated.

A higher numbers of subjects are required to be chosen enabling the authors to establish a statistical significance of the study.

6. PLOS authors have the option to publish the peer review history of their article (what does this mean?). If published, this will include your full peer review and any attached files.

Reviewer #1: No

Reviewer #2: No

---

## [Author Response · Author response to Decision Letter 0]

11 Mar 2020

Dear PLOS One editorial team and reviewers,

We appreciate your time, and the time of the reviewers, in carefully reviewing this article. In response, we have made several significant edits to the manuscript, which are detailed below. We have also taken the opportunity to update one figure (Figure 3). Please note that line numbers refer to the version of the manuscript with tracked changes.

Sincerely,

Gwenyth O. Lee

Corresponding author on behalf of the co-authors.

Reviewer Comment Response

E.1. Please ensure that your manuscript meets PLOS ONE's style requirements, including those for file naming. The PLOS ONE style templates can be found at http://www.plosone.org/attachments/PLOSOne_formatting_sample_main_body.pdf and http://www.plosone.org/attachments/PLOSOne_formatting_sample_title_authors_affiliations.pdf

We have carefully reviewed the style requirements and edited the text accordingly.

E.2 Please include a reference for the study mentioned at line 98-99, which illustrates the validation of the instrument used in the present study. We have added this reference.

E.3 We note that you have stated that you will provide repository information for your data at acceptance. Should your manuscript be accepted for publication, we will hold it until you provide the relevant accession numbers or DOIs necessary to access your data. If you wish to make changes to your Data Availability statement, please describe these changes in your cover letter and we will update your Data Availability statement to reflect the information you provide. We have already made the data available in an open public data repository as for our previous PLOS One publication 

PMC6490897 at: http://www.ifhad.org/data-repository/.

E.4 We note that you have included the phrase “data not shown” in your manuscript. Unfortunately, this does not meet our data sharing requirements. PLOS does not permit references to inaccessible data. We require that authors provide all relevant data within the paper, Supporting Information files, or in an acceptable, public repository. Please add a citation to support this phrase or upload the data that corresponds with these findings to a stable repository (such as Figshare or Dryad) and provide and URLs, DOIs, or accession numbers that may be used to access these data. Or, if the data are not a core part of the research being presented in your study, we ask that you remove the phrase that refers to these data. We have clarified this by changing “data not shown” to “model results not shown”, which is more accurate. have already made the data available in an open public data repository as for our previous PLOS One publication PMC6490897 at: http://www.ifhad.org/data-repository/

E.5 Please ensure that you refer to Figure 1 in your text as, if accepted, production will need this reference to link the reader to the figure. We have added this.

“Cough recordings were collected using a modified version of our existing cough recording device (Figure 1)” (LINE 89)

E.6 Please include captions for your Supporting Information files at the end of your manuscript, and update any in-text citations to match accordingly. Please see our Supporting Information guidelines for more information: http://journals.plos.org/plosone/s/supporting-information.

We had added captions for supporting information at the end of the manuscript.

R1.1 It is a novel presentation in which a symptom which is subjective has been expressed objectively. A mention on the calibration of the test also the specificity and sensitivity of this test needs to be addressed.

 We have added the following additional detail:

“Our previously reported algorithm then used identify potential coughs based on the sensor signal, with a sensitivity of 75.5% and a Birring specificity of 99.3% among adults. [10,11]. These recordings were then reviewed by a human listener to further increase sensitivity [14]. In a set of test recordings in which induced cough and non-cough sounds (throat clearing and spoken words) were captured by both audio and the vibration-based sensor, there was perfect agreement between classification of sounds between the two methods. A full description of this validation of the modified device will be reported elsewhere [16].” (LINES 97-103)

R1.2 Did the exclusion criteria list smokers ?

COPD is a common occurrence in smokers and can be a confounding variable for the present study.

 We had added the following additional detail, “COPD is a common occurrence in smokers, and while COPD was not noted in the clinical records of any patient, it is possible that this was an unmeasured, confounding variable.” (LINES 338-340).

R1.3 HIV patients are mentioned . Although the complications of HIV patients also need to be addressed such as pneumocystis carenii pneumonia. We have added the following additional detail, “All participants living with HIV had been previously diagnosed and were receiving antiretroviral therapy; no additional HIV-associated comorbidities, such as Pneumocystis carenii pneumonia, were noted.” (LINE 208-2010).

R2.1 The study is interesting, but the total number of subjects included in the study (n=69) is very small to reach any substantial conclusion. These numbers further reduced significantly in the sub groups, the researchers have chosen i.e. (Drug resistant n=8; HIV positive n=8; Diabetics n=6; smokers n=13) etc.

Hence, with such a small group of participants, the hypothesis, which the authors are putting forward, cannot be substantiated.

A higher numbers of subjects are required to be chosen enabling the authors to establish a statistical significance of the study. Although only 69 participants were included in the study, cough recordings, and microbiological measures of TB severity, were measured up to 6 times per each patient, therefore providing a final sample size of 358 recordings, utilizing a statistical analysis accounted for correlation between repeated measurements from the same subject. The determination of this sample size was based on estimates from prior studies (Proaño 2017 CID) and was sufficient to address two of our three study aims. While we agree with the reviewer that comparisons of cough based on drug resistance, HIV status, diabetes co-infection, and smoking, are based on a small number of participants, these numbers are also larger than those reported in other studies of cough, for example, a comparison of cough frequency and radiologic features has recently been reported with only 41 participant (Proaño 2018 Chest). Furthermore, no other report, to our knowledge, has examined the relationship between HIV, drug status, diabetes, or cough. For that reason, we consider that the associations we report, although limited, nevertheless provide value. Indeed, whether 358 recordings in 69 participants may seem subjectively small to one person or large to another, the statistical significance of our findings demonstrates that this was an adequate sample size at least for some analyses. To address the reviewer’s concern, we have clarified that the third aim of the report is exploratory, and that results should be used to guide future investigations.

This is somewhat implied by the unchanged text: 

“”The sample size was determined based on a calculation to detect differences in the proportion of positive microbiologic results between patients with and without cough and the study was not designed, a priori, to detect differences in cough between patients based on HIV co-infection or drug resistant tuberculosis.” (LINES 111-114)

In response to this suggestion, we have described the key sample sizes more explicitly in the text, so they are emphasized without readers having to refer to the Table:

“Seventy-one patients were enrolled. Sixty-nine provided at least one successful recording and were considered ‘analyzable’ cases. A total of 363 recordings were collected, of which one was unusable. 43 participants had complete data (6 recordings), 15 had 5 complete recordings, and 11 had 4 or fewer completed recordings. Among completed recordings, 358 were paired with a MODS test result from the same visit (S1 Fig 1). Characteristics of the analyzable cases are shown in Table 1. All participants living with HIV had been previously diagnosed and were receiving antiretroviral therapy; no additional HIV-associated comorbidities, such as Pneumocystis carenii pneumonia, were documented. Fifty-two recordings were available from patients who had not yet started therapy (i.e. on their day of diagnosis). The final study sample includes 69 patients with at least one paired cough recording and microbiological result., 6 of whom had diabetes co-morbidity and 7 of whom had previous TB disease.” (LINES 207-218)

We now report the AIC to 3-4 significant figures in the following text and in the table to which this text refers, because the reviewer’s comment pointed out that the previous 7 significant figures were inappropriately precise for the sample size of our study:

“Based on model fit, TOTAL TIME COUGHING was the strongest predictor of TTP (AIC=1197), followed by COUGH EPISODE FREQUENCY (AIC=1200) and AVERAGE EPISODE DURATION (AIC=1205). TOTAL TIME COUGHING was also the best predictor of MODS positivity and smear positivity (S1 Table 3). Decreases in Akaike information criterion of four or greater have been described as “significant” [27]; using this guideline TOTAL TIME COUGHING was the best cough measure compared to EPISODE FREQUENCY.” (LINES 262-269).

In response to the reviewer’s helpful suggestions, we have edited the following text in the abstract to make the sample size clearer:

“We conducted a prospective cohort of recently diagnosed ambulatory adult patients with pulmonary tuberculosis in two tertiary hospitals in Lima, Peru. Pre-treatment and five times during the first two months of treatment, a vibrometer was used to capture 4-hour recordings of involuntary cough. A total of 358 recordings from 69 participants were analyzed using a computer algorithm…. (LINES 38-40) Tuberculosis treatment response may be meaningfully assessed by objectively monitoring the time spent coughing. This measure demonstrated that cough was increased in patients with TB recurrence or co-morbid diabetes, but not because of drug resistance or HIV co-infection.” (LINES 48-49)

We have added the following text to the introduction,

” 3. As an exploratory analysis, we identify determinants of cough severity, including HIV sero-status, diabetes co-infection, and tuberculosis drug resistance.” (LINE 80)

And to the conclusion:

, “Given the small number of participants living with HIV, with diabetes co-infection, with a drug resistant infection, or presenting with other risk factors, the results we present here should be regarded as exploratory.” (LINES 335-340).

- Additional improvements

We have updated an affiliation.

Although not required by the reviewers, whilst addressing their helpful suggestions we also made the following minor improvements. 

“Department of Infectious Disease, Imperial College London, United Kingdom”

Clarified the wording of these 2 paragraphs, without changing their meaning at all:

“First, the per-episode geometric mean, indicators of the strength of the typical cough episode, were calculated as the average episode: DURATION; AMPLITUDE; and POWER. Additionally, the total hourly sum of episodes’ total: TIME; and POWER together with the COUGH EPISODE FREQUENCY per hour were calculated. Thus our parameters were:” (LINES 144-149)

“TOTAL TIME COUGHING was significantly positively associated with: diabetes (β=0.86, 95% CI: 0.09, 1.63, p=0.028); history of prior tuberculosis (β=1.44, 95% CI: 0.66, 2.22, p<0.001) but not HIV status nor drug resistant tuberculosis. Smokers with TB also tended to cough more than non-smokers with TB, although this was not statistically significant. COUGH EPISODE FREQUENCY was significantly associated with the same factors (Table 2 and Fig 3).” (LINES 285-293)

And we made two, one-word grammatical improvements.

All these proposed edits are tracked into the uploaded manuscript.

---

## [Editor Report · Decision Letter 1]

18 Mar 2020

COUGH DYNAMICS IN ADULTS RECEIVING TUBERCULOSIS TREATMENT

PONE-D-19-31712R1

Dear Dr. Lee,

We are pleased to inform you that your manuscript has been judged scientifically suitable for publication and will be formally accepted for publication once it complies with all outstanding technical requirements.

With kind regards,

HASNAIN SEYED EHTESHAM

Academic Editor

PLOS ONE

Additional Editor Comments (optional):

The Authors have revised the manuscript including reference to a critical figure added from a previous publication. At many other places supporting citations were missing which have now been added. Details of test calibrations including specificity and sensitivity of the test and complications of HIV patients have been addressed. Another important comment was small sample size for which the Authors have now provided a final sample size of 358. I now recommend this manuscript for publication.
---

## [Editor Report · Acceptance letter]

25 Mar 2020

PONE-D-19-31712R1 

Cough dynamics in adults receiving tuberculosis treatment 

Dear Dr. Lee:

I am pleased to inform you that your manuscript has been deemed suitable for publication in PLOS ONE. Congratulations! Your manuscript is now with our production department. 

With kind regards,

on behalf of

Prof HASNAIN SEYED EHTESHAM 

Academic Editor

PLOS ONE